# Photovoltaic Power Generation Prediction via Pre-Training Mamba

**Anqi Li, Bowen Su and Jay Young**
Tsinghua University
2024311645, 2024310512 and 2024210972

## Abstract

The uncertainty associated with solar photovoltaic (PV) power output (PO) is a big challenge to design,manage and implement effective demand response, and management strategies. Therefore, an accurate PV power output forecast is an utmost importance to allow seamless integration and a higher level of penetration. Although there are already many methods for predicting photovoltaic power generation, there is still room for improvement in accuracy. Therefore, we plan to use the Mamba model to pretrain on a large time-series dataset first, and then fine tune it using a photovoltaic power generation dataset, hoping to improve accuracy.

## 1 Background

Modern economies depend on reliable energy sources to support essential sectors such as agriculture, healthcare, industry, education, and environmental protection. While fossil fuels accounted for 83.1% of global energy production in 2020, their continued use presents significant problems. The combustion of fossil fuels generates large quantities of greenhouse gases, leading to air and water pollution. Additionally, the rapid depletion of these resources raises serious concerns about their long-term availability and sustainability.[5]

Therefore, in recent years, the utilization of sustainable energy has developed rapidly, and power generation methods such as wind power, photovoltaic power, and geothermal power have made rapid progress. Among them, photovoltaic power generation has been widely used due to its clean and environmentally friendly, abundant resources, and low operating costs. [1]

However, the output of photovoltaic (PV) power generation is strongly influenced by weather conditions, making it susceptible to fluctuations that can significantly impact performance. When PV power represents a substantial portion of the energy mix, these variations can cause instability within the system. As a result, accurate forecasting becomes crucial for effectively incorporating PV power into electrical grids and reducing the risks associated with its variability. In response, the past decade has seen a surge in research offering innovative and promising approaches to tackle this issue.[3]

## 2 Related work

In existing studies, various approaches have been proposed for PV power forecasting, including physical models, statistical models, and machine learning models. For physical models, such as the detailed PV model, simulate the behavior of PV cells based on fundamental physical principles. These models require detailed information about the PV system and meteorological conditions, making them computationally expensive and challenging to implement in real-time applications. And the Statistical models, such as auto-regressive integrated moving average (ARIMA) models and exponential smoothing methods, analyze historical PV power data to identify patterns and trends.[7]

Submitted to 38th Conference on Neural Information Processing Systems (NeurIPS 2024). Do not distribute.

These models are relatively simple to implement and can provide reasonable short-term forecasts. However, they may not capture complex temporal dependencies and non-stationary characteristics of PV power generation. Machine learning models, such as artificial neural networks (ANNs)[8], support vector machines (SVMs)[6], and random forests, have gained popularity in PV power forecasting due to their ability to learn complex patterns from data. ANNs, in particular, have shown promising results in capturing the non-linear relationships between meteorological variables and PV power output. However, This often require large amounts of training data and may suffer from overfitting. Recently, deep learning models have emerged as powerful tools for PV power forecasting, which is normally be divided into convolutional neural network (CNN) methods[9], recurrent neural network (RNN)methods, and hybrid models of the two, where RNN usually includes LSTM[4] and GRU[2]. These models can effectively capture long-term dependencies and temporal dynamics of PV power generation. Additionally, attention-based models[10], such as the transformer, have been explored for PV power forecasting, offering improved performance and interpretability.Despite the success of existing deep learning models, there is still room for improvement in terms of flexibility, generalization, and interpretability.

## 3  Proposed Method and Definition

Mamba, a recently proposed deep learning model, offers several advantages over traditional models that make it a compelling choice for PV power forecasting. Mamba utilizes a flexible attention mechanism that allows it to effectively capture long-range dependencies and complex temporal dynamics in PV power generation data. This flexibility is crucial for accurately predicting the diverse patterns and fluctuations observed in PV power output. And Mamba's architecture enables it to learn generalizable representations of PV power generation data, making it less susceptible to overfitting and more robust to changes in input data distribution. This generalization ability is particularly valuable for PV power forecasting, where the data can exhibit significant variability due to weather conditions and other factors. lastly, Mamba incorporates an attention mechanism that provides insights into the model's decision-making process. This interpretability is crucial for understanding the factors influencing PV power generation and gaining confidence in the model's predictions. It also facilitates model debugging and refinement. For PV Power Forecasting, While there are currently no existing studies specifically exploring the application of Mamba for PV power forecasting, the model's strengths make it a promising candidate for this task. Mamba's ability to handle long sequences, capture complex dependencies, and provide interpretability aligns well with the challenges and requirements of PV power forecasting.

To evaluate the effectiveness of Mamba, we will compare it with traditional baseline models commonly used in PV power forecasting, such as ARIMA, LSTM, and Transformer-based models. ARIMA will provide a benchmark for classical time series forecasting, while LSTM and Transformer models will serve as baselines for modern deep learning methods that handle temporal dependencies in sequence data.

So we consider a problem based on a given time series sample dataset $D_{pret}$, which covers multi-domain time series data types such as power electronics, transportation, finance, diseases, and weather. We define that within dataset $D_{pret}$, the sample $x_{1:L}^{(i)}$ represents the $i$-th univariate sequence of length $L$ in the data, where $i = 1, \ldots, M$. This sequence spans from time step 1 to $L$. Assume the length of the input sequence is $L$, and the length of the predicted output sequence is $T$. We decompose $(x_1, \ldots, x_L)$ into $M$ univariate sequences $x \in \mathbb{R}^{1 \times L}$, and each is individually fed into the Mamba model to obtain the predicted output as $(y_{L+1}, \ldots, y_{L+T})$, where $y \in \mathbb{R}^{1 \times T}$. Afterwards, the actual photovoltaic power station's power generation dataset $D_{ft}$ is used as the input for the Mamba model to perform fine-tuning, further adjusting the model's performance.

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
