# OpenReview forum: "Photovoltaic Power Generation Prediction via Pre-Training Mamba"
_tsinghua.edu.cn/THU/2024/Fall/AML — THU 2024 Fall AML Submission_

### Official Review · ~Bryan_Constantine_Sadihin1 · 2024-11-09
**Review of "Photovoltaic Power Generation Prediction via Pre-Training Mamba"**

**Rating:** 8
**Confidence:** 3

**Review:**

Strength:
1. Strong Literature Basis: The proposal is grounded with past literatures for model comparison, such as ARIMA, LSTM, and Transformer models. It does comparison all the way from statistical models and all the way to modern machine learning component.

Weakness:
1. Paper Formatting: The line numbers, and lack of spacing between paragraph, is not per common NeurIPS style paper.
2. Lack of evaluation Metrics: The proposal does not specify how the Mamba models' comparison with other models are quantified

---

### Official Review · ~Sui_Yuanpei1 · 2024-11-10
**Good use of Mamba model for PV prediction**

**Rating:** 8
**Confidence:** 4

**Review:**

This proposal seeks to improve the accuracy of photovoltaic (PV) power generation forecasting by pre-training the Mamba model on a large time-series dataset and then fine-tuning it on PV-specific data. Given the fluctuations in PV output due to weather conditions, achieving accurate predictions is essential for efficient grid integration. The proposal offers a novel approach by leveraging Mamba’s strengths in handling long-range dependencies and complex temporal data, potentially providing more reliable and interpretable forecasts for PV power generation.

Pros:
1.The use of the Mamba model, known for handling long-range dependencies, aligns well with the temporal complexity of PV forecasting.
2.By combining pre-training and fine-tuning, this approach aims to increase forecast accuracy, which is critical in managing PV fluctuations.
3.The attention mechanism in Mamba enhances the model's interpretability, which is valuable for understanding key influencing factors in PV generation.

Cons:
1.The model’s performance relies heavily on the quality of the pre-training dataset, which may limit effectiveness if PV-specific nuances are not well-captured.
2. While fine-tuning improves accuracy, there is a risk of overfitting, especially if the PV dataset is small or lacks variability.
3. While the proposal includes comparisons with classical models, additional real-world benchmarking or broader datasets could provide a more robust evaluation.
4.The feasibility of using Mamba for real-time PV forecasting is unclear, particularly regarding computational demands in large-scale applications.

---

### Official Review · ~Ruitao_Jing1 · 2024-11-12
**A Novel Forecasting Method via Mamba**

**Rating:** 9
**Confidence:** 3

**Review:**

This paper addresses the challenge of photovoltaic power forecasting using machine learning techniques, providing a thorough analysis of the limitations of established methods such as exponential smoothing, ARIMA, and SVM. The authors propose the Mamba model for time series modeling, highlighting its strengths in capturing long-range dependencies, ease of generalization, and a degree of interpretability in model decisions. The novelty of the approach and the credibility of the solution are well-established.

The paper's contribution is significant, as it offers a fresh perspective on a critical issue in renewable energy management. The authors effectively demonstrate the advantages of the Mamba model over traditional methods, making a compelling case for its adoption in the field.

For further enhancement, the paper could benefit from a detailed explanation of the evaluation metrics used when comparing the Mamba model with other models like ARIMA and LSTM. This would provide a more comprehensive understanding of the model's performance relative to existing solutions. Overall, the paper is well-researched and presents a promising method for improving the accuracy of photovoltaic power forecasting.

---

### Official Review · ~Ruowen_Zhao1 · 2024-11-12
**Review on Photovoltaic Power Generation Prediction via Pre-Training Mamba**

**Rating:** 8
**Confidence:** 4

**Review:**

**Summary**

This paper points out the challenge of photovoltaic (PV) power generation and proposes to address this challenge by developing a forecasting system based on deep learning model, Mamba.

**Strength**

+ Detailed analysis on model selection: The proposal includes a thorough analysis of model selection, offering clear reasoning for why Mamba is a suitable choice for PV power forecasting.
+ Detailed problem definition: The authors provide a well-defined problem statement, clearly articulating the challenges of PV power forecasting and providing provide a precise mathematical formulation of the problem.

**Weakness**

It is suggested that the authors should include more details on Mamba fine-tuning and describe specific evaluation metrics for comparison.

---

### Official Review · ~Zhaoxi_Li2 · 2024-11-12
**Proposal Review: Photovoltaic Power Generation Prediction Using Pre-Trained Mamba Model**

**Rating:** 8
**Confidence:** 3

**Review:**

This proposal introduces a promising approach to improving photovoltaic (PV) power generation forecasting accuracy by leveraging the Mamba deep learning model, which is pre-trained on a broad time-series dataset and fine-tuned on PV-specific data. The study is well-motivated, addressing the variability in PV output caused by weather conditions—a factor that challenges grid stability and integration of renewable energy. By choosing Mamba, which features flexible attention mechanisms for capturing complex temporal patterns, the authors aim to overcome limitations seen in traditional models like ARIMA and deep learning models such as LSTMs. The proposal is well-founded, with a clear comparative evaluation plan against existing benchmarks to assess Mamba's potential. However, the approach could benefit from additional clarity on expected improvements in interpretability and flexibility over other models, as well as further insight into anticipated challenges in generalization across varied weather data. Overall, this proposal has significant potential to advance PV power forecasting, contributing meaningfully to renewable energy management and grid stability.

---

### Official Review · ~Suraj_Joshi2 · 2024-11-12
**Review of PV Power Output Using Mamba**

**Rating:** 8
**Confidence:** 4

**Review:**

This project proposal aims to predict photovoltaic (PV) power output under various conditions, an increasingly relevant task given the rise of solar energy in power grids. The choice to use Mamba, a new deep learning model designed for sequence modeling, is apt due to its high data handling capacity. The proposal also wisely suggests transfer learning to mitigate the potential data scarcity in large-scale PV output datasets.

However, there are some uncertainties in the proposal that should be addressed:
1)The authors plan to pretrain Mamba on a large dataset, but details about this dataset are lacking. It's crucial to ensure that the pretraining dataset is similar to the final task dataset to prevent performance issues.
2)The proposal attributes the shift from fossil fuels primarily to their role in air and water pollution through greenhouse gases. However, the predominant concern with greenhouse gases should be their impact on global warming.
3)It is mentioned that Artificial Neural Networks (ANNs) are not suitable due to overfitting from limited data availability. Yet, since Mamba is a larger model, it might exacerbate the overfitting issue, which could be misleading.

---

### Official Review · ~Jiuyang_Zhou1 · 2024-11-12
**Review: Photovoltaic Power Generation Prediction via Pre-Training Mamba**

**Rating:** 10
**Confidence:** 4

**Review:**

This paper mainly explores the method of using the Mamba model for photovoltaic power generation prediction. With the rapid development of sustainable energy utilization, photovoltaic power generation is widely used due to its clean and environmentally friendly nature, abundant resources and low operating costs. However, its output is greatly affected by weather conditions and fluctuates significantly, so accurate prediction is crucial for its integration into the power grid. Existing prediction methods include physical models, statistical models and machine learning models, each with its own advantages and disadvantages. The Mamba in deep learning models has the advantages of handling long sequences, capturing complex dependencies and providing interpretability, which is suitable for photovoltaic power generation prediction. The authors plan to pre-train the Mamba model on a large time-series dataset first and then fine-tune it with a photovoltaic power generation dataset. By comparing it with traditional baseline models, the effectiveness of the model is evaluated, which is expected to improve the prediction accuracy and provides new ideas and methods for research in the field of photovoltaic power generation.

---

### Official Review · ~Kaiyuan_Zhang6 · 2024-11-12
**Good proposal**

**Rating:** 8
**Confidence:** 4

**Review:**

This paper focuses on the prediction task of photovoltaic power generation, which is meaningful since solar energy may become a worldwide power in the future. Background, related work and basic methods are proposed, with some detailed formula and implementation plans.

However, I wonder whether this machine learning tasks is over simply defined, since the PV power forecasting depends on many enviroment such as weather, humidity, air quality, refractive index, etc. A single mamba framework may be not enough to make a prediction without any expert knowledge. Hope things will work out.

---

### Official Review · ~Isak_Tønnesen1 · 2024-11-12
**Review: Photovoltaic Power Generation Prediction via Pre-Training Mamba**

**Rating:** 8
**Confidence:** 4

**Review:**

The proposal addresses photovoltaic power generation prediction using the Mamba model, combining pre-training on large time-series data with fine-tuning on PV-specific data. While the methodology is well-structured and the use of Mamba for handling temporal dependencies is promising, the proposal lacks specific details about evaluation metrics and the pre-training dataset. The problem is highly relevant given the growing importance of solar energy, but more attention to environmental factors beyond just temporal data could strengthen the approach. Despite these limitations, the proposal's novel application of Mamba to PV forecasting shows potential for improving prediction accuracy.

---

### Official Review · ~Zhu_Zhang6 · 2024-11-12
**A good attempt to integrate Mamba with practical applications**

**Rating:** 9
**Confidence:** 3

**Review:**

**Summary:**

This proposal introduces a novel approach to photovoltaic (PV) power generation forecasting by using the Mamba model. The authors argue that accurate PV power forecasting is essential due to the fluctuating nature of solar energy caused by weather conditions, impacting energy reliability and grid stability. The Mamba model, with its attention mechanisms, is proposed to handle these challenges by effectively capturing long-range dependencies and complex temporal patterns in PV data, thereby improving prediction accuracy. The model will be pretrained on a large, multi-domain time-series dataset and fine-tuned on a specific PV power dataset to maximize its forecasting capability.

**Strengths:**

1. **Novel Model Application:** The choice of the Mamba model for PV power forecasting is innovative and aligns well with the task's needs for handling complex dependencies and variability in data.
2. **Pretraining and Fine-Tuning Approach:** The strategy of pretraining on a broader time-series dataset before fine-tuning on PV data may enhance the model's generalizability and robustness against overfitting.

**Weaknesses:**

1. **Limited Baseline Comparisons:** The proposal lacks sufficient details on how the Mamba model will be compared to other recent deep learning models, especially those that have shown success in PV forecasting.

**Questions:**

1. How will the authors handle the variability in weather conditions that might lead to large fluctuations in PV power output?
2. What specific evaluation metrics will be used to measure the improvement of Mamba over other baseline models?

---

### Official Review · ~Kuanghao_Wang1 · 2024-11-12
**Great idea**

**Rating:** 8
**Confidence:** 4

**Review:**

This paper focuses on forecasting photovoltaic (PV) electricity forecasts using mamba models. Uncertainty in photovoltaic (PV) power output poses a challenge in designing, managing, and implementing effective quantity on demand (QOD) response strategies. This project plans to use the Mamba model to first pre-train it on a large time series dataset and then fine-tune it using a PV electricity generation dataset to improve forecast accuracy. This paper has a detailed background and literature research and gives a mathematical definition of the prediction set. However, the paper does not specifically give the superiority of the mamba model over other models and the problem is simplified, which may not lead to more accurate predictions.